# Immunotherapy and Modern Radiotherapy Technique for Older Patients with Locally Advanced Head and Neck Cancer: A Proposed Paradigm by the International Geriatric Radiotherapy Group

**DOI:** 10.3390/cancers14215285

**Published:** 2022-10-27

**Authors:** Nam P. Nguyen, Lyndon Kim, Juliette Thariat, Brigitta G. Baumert, Thandeka Mazibuko, Olena Gorobets, Vincent Vinh-Hung, Huan Giap, Tahir Mehmood, Felix Vincent, Alexander Chi, Trinanjan Basu, Gokoulakrichenane Loganadane, Mohammad Mohammadianpanah, Ulf Karlsson, Eromosele Oboite, Joan Oboite, Ahmed Ali, Brandi R. Page

**Affiliations:** 1Department of Radiation Oncology, Howard University, 2041 Georgia Ave NW, Washington, DC 20060, USA; 2Department of Neurology, Division of Neuro-Oncology, Mount Sinai Hospital, New York, NY 10029, USA; 3Francois Baclesse Cancer Center, 14000 Caen, France; 4Institute of Radiation Oncology, Cantonal Hospital Graubuenden, 7000 Chur, Switzerland; 5International Geriatric Radiotherapy Group, Department of Radiation Oncology, Washington, DC 20001, USA; 6Department of Maxillofacial Surgery, Centre Hospitalier Universitaire de Martinique, 97213 Le Lamentin Martinique, France; 7Department of Radiation Oncology, Centre Hospitalier de la Polynesie Francaise, 98716 Pirae, Tahiti, French Polynesia; 8Department of Radiation Oncology, Medical University of South Carolina, Charleston, SC 29425, USA; 9Department of Radiation Oncology, Northampton General Hospital, Northampton NN1 5BD, UK; 10Department of Surgery, Southern Regional Health System-Lawrenceburg, Lawrenceburg, TN 38464, USA; 11Department of Radiation Oncology, Beijing Chest Hospital, Beijing 101149, China; 12Department of Radiation Oncology, HCG Cancer Center Borivali, and HCG ICS, Mumbai, Maharashtra 400092, India; 13Department of Radiation Oncology, CHU Mondor, 94000 Creteil, France; 14Department of Radiation Oncology, Shiraz University of Medical Sciences, Shiraz 71348-14336, Iran; 15Department of Hematology Oncology, Howard University, Washington, DC 20059, USA; 16Department of Radiation Oncology, Johns Hopkins University, Baltimore, MD 21093, USA

**Keywords:** older, cancer patients, locally advanced, TPS, CPI, IGRT

## Abstract

**Simple Summary:**

Immunotherapy with checkpoint inhibitors (CPI) is well tolerated in older cancer patients due to its safety profile. In selected patients with a high program death-ligand 1 (PD-L1) tumor expression defined as 50% or above, the response rate and survival are significantly better than those for conventional chemotherapy. Modern radiotherapy techniques, such as intensity-modulated image-guided radiotherapy (IM-IGRT), volumetric modulated arc therapy (VMAT) or proton therapy, enhance the tumor’s response to CPI while sparing the normal organs from excessive irradiation. Thus, the combination of IM-IGRT and CPI should work well in older head and neck cancer patients with a high PD-L1 expression who are not candidates for cisplatin-based chemotherapy due to pre-existing comorbidities. This hypothesis should be tested in future prospective clinical trials.

**Abstract:**

The standard of care for locally advanced head and neck cancer is concurrent chemoradiation or postoperative irradiation with or without chemotherapy. Surgery may not be an option for older patients (70 years old or above) due to multiple co-morbidities and frailty. Additionally, the standard chemotherapy of cisplatin may not be ideal for those patients due to oto- and nephrotoxicity. Though carboplatin is a reasonable alternative for cisplatin in patients with a pre-existing hearing deficit or renal dysfunction, its efficacy may be inferior to cisplatin for head and neck cancer. In addition, concurrent chemoradiation is frequently associated with grade 3–4 mucositis and hematologic toxicity leading to poor tolerance among older cancer patients. Thus, a new algorithm needs to be developed to provide optimal local control while minimizing toxicity for this vulnerable group of patients. Recently, immunotherapy with check point inhibitors (CPI) has attracted much attention due to the high prevalence of program death-ligand 1 (PD-L1) in head and neck cancer. In patients with recurrent or metastatic head and neck cancer refractory to cisplatin-based chemotherapy, CPI has proven to be superior to conventional chemotherapy for salvage. Those with a high PD-L1 expression defined as 50% or above or a high tumor proportion score (TPS) may have an excellent response to CPI. This selected group of patients may be candidates for CPI combined with modern radiotherapy techniques, such as intensity-modulated image-guided radiotherapy (IM-IGRT), volumetric arc therapy (VMAT) or proton therapy if available, which allow for the sparing of critical structures, such as the salivary glands, oral cavity, cochlea, larynx and pharyngeal muscles, to improve the patients’ quality of life. In addition, normal organs that are frequently sensitive to immunotherapy, such as the thyroid and lungs, are spared with modern radiotherapy techniques. In fit or carefully selected frail patients, a hypofractionated schedule may be considered to reduce the need for daily transportation. We propose a protocol combining CPI and modern radiotherapy techniques for older patients with locally advanced head and neck cancer who are not eligible for cisplatin-based chemotherapy and have a high TPS. Prospective studies should be performed to verify this hypothesis.

## 1. Introduction

The standard of care for locally advanced head and neck cancer includes postoperative irradiation or concurrent chemotherapy and irradiation [1,2]. However, older cancer patients defined as 70 years old or older present a particular challenge. The prevalence of comorbidities increases with age [3]. Older head and neck cancer patients in particular experience a high rate of co-morbidity, which further increases following treatment likely due to a past medical history of heavy smoking and drinking [4]. Serious complications and the mortality rate increase significantly with age among older head and neck cancer patients who undergo surgery [5]. Frailty, defined as an accumulative decline in physical reserve, is a serious issue which affects one’s tolerance to head and neck cancer chemoradiation, independent of age. As an illustration, among 502 patients aged 24 to 60 years who underwent concurrent chemoradiation for locally advanced head and neck cancer, 33.7% were frail. Those patients had a significantly higher rate of hospitalization, emergency room visits, and treatment incompletion compared to the fit ones [6]. Thus, it is not surprising that among older patients who are deemed fit enough to undergo concurrent chemoradiation for locally advanced disease, high rates of treatment discontinuation, grade 3–4 toxicity and poor survival have been reported [7,8]. In addition, a significant number of older head and neck cancer patients receive substandard treatment based on their chronological age, likely due to physicians’ fear of treatment toxicity, resulting in its discontinuation [9,10,11]. Among patients who are able to complete treatment, serious complications induced by standard cisplatin-based chemotherapy include oto- and nephtotoxicity, which may decrease their quality of life after treatment [12,13,14]. Even though carboplatin is a reasonable alternative for patients with a pre-existing hearing deficit or renal insufficiency, its efficacy may be inferior to cisplatin for head and neck cancer [15,16]. When combined with another chemotherapy agent, such as 5-fluorouracil (5-FU), for head and neck cancer irradiation, there was more treatment toxicity and interruption in patients who had received the two chemotherapy agents compared to those receiving cisplatin alone [17]. Older cancer patients also have a reduced bone marrow reserve, which may lead to prolonged anemia, neutropenia, lymphopenia and an increased risk of infection and delayed recovery [18,19,20]. Previous attempts to replace cisplatin with a lesser toxic agent, such as cetuximab, to improve patient tolerance to systemic treatment have resulted in a poorer survival rate [21,22]. In addition, a hypofractionated regimen combining cetuximab with radiotherapy to reduce the treatment time may result in excessive toxicity among older and frail head and neck cancer patients [23]. Thus, clinicians should devise a new algorithm to treat this vulnerable older cancer population to optimize their treatment while minimizing treatment toxicity.

Recently, immunotherapy with check point inhibitors (CPI) has emerged as a promising agent for head and neck cancer patients because of its different toxicity profile and its efficacy for patients with recurrent or metastatic cancer compared to salvage chemotherapy [24,25]. As an international organization devoted to the care of older cancer patients, minorities and women who are frequently excluded from current clinical trials, the International Geriatric Radiotherapy Group (http://www.igrg.org accessed on 15 September 2022) would like to investigate if immunotherapy combined with modern radiotherapy techniques, such as intensity-modulated image-guided radiotherapy (IM-IGRT), may improve local control and/or quality of life for head and neck cancer patients who are not eligible for cisplatin-based chemotherapy [26,27]. Previous data have reported the benefit of intensity-modulated radiotherapy (IMRT) over the three-dimensional conformal radiotherapy technique in terms of organ sparing in head and neck cancer for both genders [28,29]. As currently there are controversies about gender’s effect on the outcome of head and neck cancer, any investigation needs to take into consideration the biological differences between men and women, which may potentially affect the response to treatment.

## 2. Prevalence of Program Death-Ligand 1 (Pd-L1) Receptors in Patients with Head and Neck Cancer

Among all types of cancer, it is likely that head and neck cancer presents with one of the best opportunities for CPI treatment. Program death-ligand 1 (PD-L1) is an immune check point inhibitor, which is present on all normal cells to prevent autoimmunity and also on tumor cells to escape destruction by CD8 T cells. It is highly expressed in head and neck cancer. Compared to other tumors, such as those of bladder cancer, renal cell carcinoma, lung cancer and melanoma, there is a 60-fold increase in PD-L1 expression in head and neck cancer [30]. For example, nasopharyngeal cancer has the third highest expression of PD-L1 after thymic cancer and diffuse large cell lymphoma [31]. Even though PD-L1 is not a perfect biomarker for immunotherapy, high PD-L1 expression, defined as 50% or more or a high tumor proportion score (TPS) or combined positive score (CPS), frequently indicates a high response to CPI. A high TPS has been correlated with an excellent survival, independent of tumor histology following immunotherapy [32,33]. As an illustration, in locally advanced or metastatic non-small-cell lung cancer patients with a high TPS, the rate of survival with pembrolizumab alone (*n* = 637) was superior to that of conventional chemotherapy (*n* = 637). The median survival was 20 months and 12.2 months for pembrolizumab and chemotherapy, respectively. Another benefit of immunotherapy was a reduction in toxicity. The grade 3–4 side-effects were 18% and 41% for pembrolizumab and chemotherapy, respectively [34]. Other studies also corroborated the superior survival rate of pembrolizumab alone and a significant reduction in severe side effects for this patient population compared to that of chemotherapy in other cancers, such as esophageal cancer and uroepithelial cancer [35,36]. Thus, CPI may be a good alternative for selective older cancer patients because of its efficacy and safety profile. However, the pharmacokinetics of CPI may be different for older patients and their tolerance to the medication needs to be assessed.

## 3. Efficacy and Tolerance of Older Cancer Patients to CPI

The tolerance and efficacy of older cancer patients to CPI has been investigated in many prospective and retrospective studies with a wide range of malignancies. In a prospective longitudinal study to assess the tolerance of NSCLC and melanoma patients 70+ years of age (*n* = 70) and younger patients (*n* = 70) to various CPI treatments, there was no difference in grade 3–5 toxicity between those two groups despite a higher comorbidity index for the older patients [37]. The grade 3–5 toxicity was 18.6% and 12.9% for the older and younger patients, respectively. Other studies have also corroborated the safety of CPI for older cancer patients. Among 245 NSCLC patients from various age groups ranging from less than 60 to more than 80 who received treatments with CPI, there was no difference in toxicity between those groups [38]. The safety of CPI is also confirmed through real-world data. In a group of 197 patients with various malignancies ranging from head and neck cancer to renal cell carcinoma treated with different types of CPI, the grade 3–5 toxicity was 48.3% and 37.4% (*p* = 0.1) for patients 75 years old or above (*n* = 58) and less than 75 (*n* = 139), respectively [39]. The response rate to CPI was also similar to the one reported in the literature for older cancer patients [40]. A meta-analysis of 11,157 patients included in 19 randomized CPI trials reported an improved survival rate and progression-free survival for both older and younger cancer patients compared to those of the control group [41]. However, the survival benefit of CPI was greater for older patients (65 years old or above) for each cancer histology, which raises interesting issues for future prospective studies. Another meta-analysis has also corroborated the survival benefit of CPI for older cancer patients [42].

## 4. Potential Advantages of CPI Compared to Cisplatin-Based Chemotherapy

### 4.1. Ototoxicity

As patients get older, their hearing declines progressively. According to the National Institute of Aging (NIA) (https://www.nia.nih.gov/health accessed on 15 September 2022), one out of three seniors older than 60 have hearing loss. By the time they reach 85 or older, one out of two will develop a hearing deficit which significantly decreases their quality of life (QOL) due to the difficulty communicating effectively and social isolation [43]. Thus, any treatment which potentially increases hearing loss should be avoided to prevent the further deterioration of patients’ QOL. Cisplatin, by virtue of its ototoxicity, should not be used for older cancer patients with a pre-existing hearing deficit if an alternative effective treatment is available. In addition, conventional radiotherapy may further compound the severity of hearing loss due to its apoptotic effect on cochlea cells which is dose-dependent. The combination with a radiosensitizing agent during irradiation may decrease the threshold for radiation damage, as the cochlea is an organ at risk (OAR) close to the planning target volume (PTV) [44]. Indeed, a prospective study comparing radiotherapy alone to concurrent chemoradiation for patients with nasopharyngeal cancer has reported a significant increase in hearing loss for the combined treatment group [45]. Loss of hearing occurs during or after cisplatin-based chemoradiation and increases over time after treatment [46,47].

On the other hand, ototoxicity is a very rare event which may occur during CPI administration and is related to autoimmune inner ear disease [48,49]. In contrast to cisplatin-induced ototoxicity which is frequently irreversible, intratympanic steroid injection and/or the discontinuation of immunotherapy may improve the severity of the hearing loss.

### 4.2. Nephrotoxicity

The prevalence of renal failure increases in the elderly population, and ranges from 7 to 55% in patients 60 years old or above [50]. According to the Centers for Disease Control and Prevention (https://www.cdc.gov/kidneydisease accessed on 15 September 2022), the prevalence of chronic kidney disease is 38% among people aged 65 or older due to a higher rate of comorbidity, such as high blood pressure, diabetes and vascular disease [51]. Thus, those affected may not be candidates for cisplatin-based chemotherapy, which may be replaced by carboplatin or cetuximab. However, those agents may not be optimal to improve survival compared to cisplatin [16,21,22].

Immunotherapy can also cause acute renal failure through an immune mechanism. Acute interstitial nephritis, acute tubular injury and minimal change disease have been reported [52,53]. Those complications are also relatively rare (1.7%) and reversible with systemic steroid administration and drug withdrawal.

### 4.3. Bone Marrow Toxicity

Cisplatin can cause severe grade 3–4 hematologic toxicity through its direct effect on the bone marrow [54]. In addition, prolonged anemia can persist after treatment because of decreased erythropoietin production on the renal tubules [55]. Older patients are particularly prone to chemotherapy-induced bone marrow suppression due to the aging of hematopoetic stem cells [18]. Medications that are less toxic to the bone marrow would be preferable if equally effective.

Even though hematologic toxicity has been reported with immunotherapy, and may result in death if unrecognized, its incidence remains low compared to that of systemic chemotherapy. A review of the World Health Organization’s pharmacovigilance database reported only 168 cases of CPI-induced hematologic toxicity out of a database of 16 million patients who received the drugs [56]. Clinical data has also supported the efficacy and the superior hematologic profile of CPI compared to cisplatin-based chemotherapy. In patients with advanced gastric cancer and a positive PD-L1 expression, the three-year survival rate was 60% and 20% for pembrolizumab alone (*n* = 256) and chemotherapy (*n* = 250), respectively. The corresponding grade 3–5 toxicity and hematologic toxicity were 16.9 and 69.3%, and 2.4 and 31.1%, respectively [57]. Other randomized studies also corroborated the reduced hematologic toxicity of immunotherapy in patients with a high TPS score in NSCLC and bladder cancer [34,58]. The grade 3–5 hematologic toxicity was 4 and 41%, and 12 and 71% for NSCLC and uroepithelial carcinoma treated with immunotherapy and chemotherapy, respectively. However, only the NSCLC patients had superior survival rates with immunotherapy alone [57]. Thus, the treatment with CPI alone may reduce serious toxicity and improve survival in the selected group of patients.

## 5. Potential Disadvantages of CPI Compared to Cisplatin-Based Chemotherapy

The potential benefit of CPI may be negated by its impact on the endocrine, gastrointestinal and cardiac systems among others in older cancer patients. Misdiagnoses may lead to death. However, it is important to note that there was no significant difference in the grade 3–5 toxicity between fit and frail older cancer patients undergoing immunotherapy. Among 92 patients 70 years of age or older who received CPI for metastatic melanoma, the prevalence of grade 3–5 was 17% and 27% for the fit and frail patients, respectively [59]. Other studies have corroborated the safety of CPI among older cancer patients who experienced various grades of frailty ranging from low to severe [39,60]. Even though the data are only preliminary and involve a small number of patients, they suggest that frailty is not an absolute contraindication for immunotherapy in older cancer patients. However, those patients should be monitored carefully during treatment to avoid serious complications. It is important to note that there are no clear guidelines on how to monitor older cancer patients who may have multiple comorbidities and various degrees of frailty. The symptoms and signs of complications which may occur as a side effect of immunotherapy, such as myocardial infarction, may not be specific in older women, for example [61,62]. Thus, a great deal of vigilance is required from clinicians during treatment, and when in doubt, patient referral to specialty services is recommended. Immunotherapy has been reported to cause complications due to the excessive immune response of normal organs. Hypophysitis, thyroid dysfunction, type I diabetes and adrenal insuffiency have been reported [63,64]. A combination of CPI treatments may lead to a higher risk of endocrine toxicity compared single-agent CPI [64].

Among the immune side effects and complications that may occur during head and neck cancer treatment, hypophysitis may be the most challenging due to its insidious clinical presentation with serious consequences, such as adrenal insufficiency, if undiagnosed [65]. At a minimum, morning pituitary and target hormones’ baseline levels, including adrenocorticotropic hormone (ACTH), thyroid-stimulating hormone (TSH), follicle-stimulating hormone (FSH), luteinizing hormone (LH), growth hormone (GH), prolactin, cortisol, free thyroid hormones, testosterone, estradiol, progesterone and insulin-like growth factor-1 (IGF-1), should be drawn before treatment and repeated if clinically indicated. Special testing and referral to the endocrinology service may be needed to diagnose pituitary insufficiency. Hormonal replacement should be initiated in that case.

Another debilitating side effect which occurs frequently during immunotherapy with devastating consequences for older patients is colitis. Diarrhea with or without abdominal pain could indicate an early development of bowel inflammation. Depending on the diarrhea severity according to the National Cancer Institute’s (NCI) common terminology criteria for adverse events (CTCAE), hospital admission, an endoscopic exam, and a biopsy may be required [66,67]. As the left colon is most commonly involved, a flexible sigmoidoscopy is initially the test of choice to minimize the risk of colonic dilatation and perforation. A biopsy of the rectum and left colon is routine, even if the mucosa appears normal to document histologic inflammation. Grade 3 toxicity requires the discontinuation of immunotherapy and the initiation of a high dose of corticosteroid, and in refractory cases, immunosuppressive therapy [68].

## 6. Efficacy of CPI in Patients with Recurrent Head and Neck Cancer

Recurrent head and neck cancer carries a poor prognosis with a median survival rate ranging from one to 13 months after salvage with chemotherapy [69]. Thus, investigations for newer systemic agents have been conducted to improve the outcome of those patients. Phase I–II studies of CPI for recurrent or metastatic head and neck cancer refractory to cisplatin and cetuximab have been promising with a high response rate and acceptable toxicity [70,71,72]. Randomized studies have corroborated the efficacy and safety of CPI as a salvage agent for recurrent or metastatic head and neck cancer [24,25,73]. Pembrolizumab alone (*n* = 247) was randomized to the investigators’ choice of methotrexate, docetaxel, or cetuximab (*n* = 248) for salvage therapy in platinum-resistant head and neck cancer patients. The median survival was 8.4 months and 6.9 months for the pembrolizumab and standard care groups, respectively. Grade 3–5 toxicity was also significantly less for pembrolizumab (13%) compared to that of the standard care group (18%). Later on, pembrolizumab alone (*n* = 301) was proven to be superior to the combination of cetuximab and chemotherapy (*n* = 300) for recurrent or metastatic head and neck cancer patients [25]. Pembrolizumab’s survival benefit was greatest among patients with a high TPS. The median survival was 14.9 months and 10.7 months for the pembrolizumab alone and cetuximab with chemotherapy groups, respectively. Grade 3–5 toxicity was 55% and 83% for pembrolizumab alone and cetuximab with chemotherapy, respectively. Thus, pembrolizumab alone has been approved for recurrent or metastatic head and neck cancer with a high TPS. Another CPI, nivolumab, has also been demonstrated to be superior to the standard care of single-agent chemotherapy [74]. The median survival was 7.5 months and 5.1 months for nivolumab alone (*n* = 240) and standard therapy (*n* = 121), respectively. The corresponding grade 3–4 toxicity was 13.1% and 35.1%, respectively. The survival benefit of nivolumab was maintained after long-term follow-ups [74]. Interestingly, the survival benefit of nivolumab was observed regardless of age [75]. The 30-month survival rates of patients 65 years of age or older were 13% and 3.3% for nivolumab and the standard care therapy group, respectively. Thus, CPI alone may be beneficial for older head and neck cancer patients and may be potentially combined with modern radiotherapy techniques, such as IM-IGRT, to reduce treatment toxicity. The reported response to immunotherapy is about 20% for all head and neck cancer patients [76]. Table 1 summarizes the benefit of immunotherapy over conventional chemotherapy for the selected patients.

## 7. Intensity-Modulated Image-Guided Radiotherapy for Older Cancer Patients

The search for new radiotherapy techniques to reduce treatment toxicity has pushed radiation oncologists to evolve from two-dimensional (2-D) and three-dimensional (3-D) conformal radiotherapies and IMRT to IMRT-IGRT. Daily cone-beam computed tomography (CBCT) or other imaging techniques, such as magnetic resonance imaging (MRI), before treatment allows for an accurate visualization of the tumor and patient positioning to avoid a marginal miss while minimizing damage to the organs at risk (OARs) surrounding the tumor [77]. Daily imaging is particularly critical in the head and neck region due to the close proximity of multiple OARs which are very sensitive to radiation damage, such as the cochlea, parotids, larynx, pharyngeal muscles, mandible and spinal cord [78]. Hearing loss, xerostomia, hoarseness of the voice, dysphagia and in severe cases, aspiration and osteroradionecrosis are serious complications which impact patients’ QOL following chemoradiation for head and neck cancer [79]. Aspiration during head and neck cancer irradiation may result in death from aspiration pneumonia or a prolonged feeding tube which significantly impairs patients’ QOL [80,81]. Those complications may not occur if the threshold for radiation damage to the OAR is not exceeded. Compared to alternative radiotherapy techniques, IM-IGRT allows a significant reduction in radiation dose to those OAR which may reduce complication rates [82,83,84,85]. The severity of lymphopenia during head and neck cancer radiotherapy may also be potentially reduced due to a lower radiation dose to the target area at a low risk for recurrence [20].

As a result of the normal tissue OAR-sparing properties of IM-IGRT, older head and neck cancer patients are more amenable to full-course chemoradiation. In one such study about IM-IGRT, there was no difference in the grade 3–4 toxicity or survival rates between younger (less than 70 years old) or older (70 years old or above) patients [27]. The preliminary results on patients’ QOL following head and neck cancer for IM-IGRT are also promising. Hoarseness of the voice is significantly reduced because of the decreased laryngeal edema secondary to laryngeal sparing [86]. Another study reported up to 70% of the head and neck cancer patients experienced excellent QOL one year following treatment [87]. Thus, IM-IGRT may improve further patients’ QOL if combined with immunotherapy, as grade 3–4 hematologic toxicity remains significant with cisplatin-based chemotherapy regardless of age. The combination of immunotherapy and radiotherapy is well tolerated and provides excellent local control for other cancers that are traditionally radioresistant, such melanoma and renal cell carcinoma, because of the synergy between those two modalities and it may serve as a template for selected head and neck cancer patients [88,89].

Hypofractionated radiotherapy has been proposed as an effective technique to reduce the treatment time for head and neck cancer patients. However, in older and frail patients with locally advanced head and neck cancer, the combination of cetuximab and hypofractionated radiotherapy has been reported to be associated with excessive grade 3 and 4 toxicity [23]. It is likely that severe mucositis induced by cetuximab may be responsible for the poor tolerance of those patients to irradiation [90]. In contrast, immunotherapy with CPI is well tolerated even among frail older cancer patients. Thus, fit older cancer patients should do well with hypofractionated radiotherapy and CPI. Given the safety of CPI’s profile, special consideration should be given for concurrent immunotherapy and hypofractionated IM-IGRT or other normal organ-sparing techniques, such as proton therapy for older and frail head and neck cancer patients. Those patients should be monitored carefully during treatment to assess any adverse events.

A review of the literature has reported that hypofractionated radiotherapy is safe and effective for older head and neck cancer patients, either with a palliative or curative intent [91]. Preliminary data of concurrent chemotherapy and hypofractionated IMRT for locally advanced head and neck cancer have shown excellent local control with an acceptable level of toxicity [92,93]. A randomized study comparing concurrent chemotherapy with hypofractionated versus conventional fractionated IMRT has reported no difference in the survival or toxicity between these two branches [94]. Even though the number of patients is small, this study demonstrated that hypofractionated IMRT may be cost-effective and may improve older patients’ QOL by reducing the need for transportation. Thus, hypofractionated IMRT-IGRT may be a better option for older patients with locally advanced head and neck cancer rather than a conventional fractionation or a split treatment course to limit treatment toxicity [95].

## 8. The Potential Role of Proton Therapy for Older Cancer Patients 

Proton therapy has been proposed as a potential radiotherapy modality to improve older head and neck cancer patients’ tolerance to treatment due to its normal organ-sparing property [96]. Indeed, the limited penetration of protons associated with its Bragg peak effect allows proton therapy to deliver a high radiation dose to the tumor and for the sparing of normal organs [97]. However, the cost and limited availability of proton centers make it impractical for the general cancer population [98]. When available, in some head and neck cancers, such as nasopharyngeal cancer, the use of proton radiotherapy may improve oral mucositis rates associated with lower feeding tube placement rates; investigations are ongoing [99]. For some salivary gland tumors treated with ipsilateral neck irradiation alone, proton therapy may confer a benefit to reducing oral mucositis by decreasing the secondary dose spillage [100].

## 9. The Potential Role of Volumetric Modulated Arc Therapy for Older Cancer Patients

The introduction of volumetric modulated arc therapy (VMAT) has significantly improved head and neck cancer OAR sparing. A combination of three parameters during treatment, such as gantry rotation speed, treatment aperture shape and dose rate, allows for an effective radiotherapy delivery with less monitored units within a shorter period of time compared to IM-IGRT [101,102]. However, planning for VMAT requires a significantly longer period of time than IM-IGRT [101]. Preliminary results of VMAT for head and neck cancer treatment are very encouraging. An improved overall survival rate and a reduction in acute and late grade 3–4 toxicity were reported and attributed to the normal organ-sparing property of VMAT [103,104,105,106]. Older head and neck cancer patients also tolerated their treatment with VMAT very well compared to younger patients [107]. Thus, VMAT is another treatment modality which may be combined with immunotherapy for older head and neck cancer patients with locally advanced diseases. Table 2 summarizes the advantages and limitations of these radiotherapy techniques.

## 10. The Potential Role of Gender on Survival Outcome in Head and Neck Cancer

There is currently a controversy on the role of gender in the survival outcome of patients with head and neck cancer. Many retrospective studies have reported that the survival of women with head and neck cancer may be better compared to males with the same stage [108,109,110]. As an illustration, women with laryngeal cancer may have a lower recurrence rate compared to men with the same disease stage [108,109,110]. As an illustration, women with laryngeal cancer may have a lower recurrence rate compared to men following treatment but may experience a higher risk of secondary malignancies [109]. The biologic milieu of women with a preponderance of estrogen may have accounted for the survival difference between men and women with head and neck cancer [111]. In mice with implanted human-papillomavirus-positive tumor cells, the responses of female mice to chemotherapy and immunotherapy were significantly better compared to those of males [112]. Thus, it was postulated that the estrogen tumor environment may have favorably influenced the treatment response. Clinical studies are in agreement with this experiment. For example, chemotherapy and immunotherapy agents have a longer half-life in women due to the difference in pharmacokinetics, resulting in a better response and/or an increase in toxicity [113]. In head and neck cancer patients receiving 5-fluorouracil-based induction chemotherapy, among patients who developed a complete response, the plasma concentration in women was higher than that of men [114]. In a meta-analysis of head and neck cancer patients receiving chemotherapy, women benefited from the treatment more than men [115]. Another meta-analysis also corroborates a favorable response of women to anti-PD-L1-based therapy and raises concerns about the potential role of gender in influencing survival outcome [116]. However, using a match-paired analysis to compare survival rates between men and women following head and neck cancer treatment, there was no gender benefit [117]. Thus, the role of gender in influencing treatment outcome in head and neck cancer is currently unclear. As women are currently under-represented in head and neck cancer clinical trials and half of the reported studies did not analyze sex as an independent variable, this controversy remains unanswered [118,119].

## 11. Proposed Algorithm for Older Patients with Locally Advanced Head and Neck Cancer

We propose that older locally advanced head and neck cancer patients (65 years old or above) with a high TPS who are not eligible for cisplatin-based chemotherapy should be enrolled in a protocol of concurrent immunotherapy and IM-IGRT, VMAT or proton therapy if available. The efficacy, toxicity and impact of the combined treatment on patients’ QOL should be assessed. The data obtained may guide clinicians to conduct further randomized studies to test the treatment efficacy among different ethnic groups, as current clinical trials are biased toward Caucasians [120]. Among different ethnic groups in the United States, African Americans with head and neck cancer have the worst survival, independent of the stage and treatment [121,122]. Thus, further investigations need to be conducted to assess the influence of ethnicity and gender on head and neck cancer survival and require the recruitment of a large number of patients. As our International Geriatric Radiotherapy Group (IGRG) network includes over 1100 institutions in 127 countries, recruitment may not be an issue [26,123]. We acknowledge that patients with a lower TPS may still benefit from immunotherapy, but the preliminary data obtained from our studies may pave the way for further investigations [34]. Any treatment protocol for older head and neck cancer patients should be flexible enough to accommodate the special needs of this vulnerable population. A six-week immunotherapy administration interval may be preferable to minimize transportation issues for older cancer patients.

Immunotherapy can be started during or prior to radiotherapy if there is any delay, due to dental work for example. An acceptable hypofractionated radiotherapy schedule could be 6250 cGy in 250 cGy/fraction to the gross tumor volume and metastatic lymph nodes, 5500 cGy in 220 cGy/fraction to the high-risk areas, and 5000 cGy in 200 cGy/fraction to the low-risk areas [124]. Treatments could be completed within five weeks instead of the conventional seven weeks of radiotherapy.

## 12. Conclusions

Immunotherapy combined with IM-IGRT or other modern radiotherapy techniques may be beneficial for older patients with locally advanced head and neck cancer with a high TPS who are not eligible for cisplatin-based chemotherapy. The superior toxicity profile of CPI provides synergy to the organ-sparing irradiation technique, which may improve local control and survival while minimizing toxicity in this vulnerable population. Prospective studies should be conducted in the future to verify this hypothesis.

## Figures and Tables

**Table 1 cancers-14-05285-t001:** Comparison of immunotherapy and conventional chemotherapy for head and neck cancer patients.

	Immunotherapy	Chemotherapy
Advantages	superior survival for selected patients with PD-L1 expression 50% or higher less grade 3–5 toxicity	less expensive
Disadvantages	cost	toxicity especially bone marrow toxicity and mucositis

PD-L1: program death ligand 1.

**Table 2 cancers-14-05285-t002:** Comparison of modern radiotherapy techniques which may be combined with immunotherapy for normal organs sparing.

	IM-IGRT	VMAT	VMAT
Advantages	Good organ sparing Short planning time Available in most centers	Good organ sparing Less MU delivered Short treatment time Available in most centers	Excellent organ sparing
Limitations	More MU delivered Longer treatment time	Longer planning time	High cost Limited in selected centers

IMRT-IGRT: intensity-modulated image-guided radiotherapy; VMAT: volumetric modulated arc therapy; MU: monitored units.

## Data Availability

Not applicable.

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
