# Peer review of "Immunotherapy and Modern Radiotherapy Technique for Older Patients with Locally Advanced Head and Neck Cancer: A Proposed Paradigm by the International Geriatric Radiotherapy Group"

_cancers, 2022, doi:10.3390/cancers14215285_

Round 1

Reviewer 1 Report

General comments:

This article proposes a change in treatment approach for older HNC patients to reduce normal tissue toxicity and to increase quality of life. The subject is of high interest to the scientific community and is compiled by the International Geriatric Radiotherapy Group with the potential to be trialled in prospective clinical studies.

The paper is well structured, and the aim of the proposal is well justified. It also fits the topic of the special issue.

I have a number of recommendations to improve the readability of the manuscript:

1.     In the Introduction (last paragraph) there is specific mention of the inclusion of women patients in studies conducted by the International Geriatric Radiotherapy Group. Please comment on the gender-specific radiotherapy / chemotherapy (particularly platinum-based chemo) / immunotherapy that should be taken into account when designing a clinical trial for older patients, especially since the aim is to reduce unwanted side effects. Similar comment for section 9. Proposed algorithm for older patients …

2.     While the authors focus on intensity modulated radiotherapy, they should mention the role of VMAT in HNC normal tissue sparing, with similar (if not superior) benefits as IMRT. There are several recent articles on dosimetric comparison between IMRT and VMAT in HNC evaluating the potential benefits of VMAT in this patient group. Section 7 of the paper should include a paragraph specifying the prospect of using VMAT (and not only IMRT) as intensity modulated irradiation technique. VMAT can further reduce the dose to critical organs due to lower monitor units administered as compared to IMRT.

3.     The paper must undergo a thorough English language revision as there are several typos and grammatically incorrect sentences throughout the paper (some depicted in the Specific comments).

4.     The authors should evaluate if the large number of auto-citations is really necessary.

Specific comments:

1.     Abstract: “Additionally, the standard chemotherapy of cisplatin which may not be ideal for those patients due to oto- and nephrotoxicity.” Please rephrase, the sentence is not grammatically correct (or remove ‘which’).

2.     Abstract: “In addition, concurrent chemoradiation is frequently associated with grade 3-4 mucositis and hematologic toxicity for which older cancer patients with borderline physical function may not tolerate.” Please rephrase, the sentence is not grammatically correct.

3.     Line 102 – ‘…compared to salvage chemotherapy.’

4.     Line 129 – ‘..toxicity was also significantly reduced…’

5.     Line 226 – replace ‘countraindication’ with ‘contraindication’

6.     Lines 233-234 – rephrase the sentence starting with ‘Many adverse events …’

7.     Line 237 – ‘…compared to single agent..’

8.     Line 252 – remove ‘criteria’ after CTCAE – is repetitive

9.     Line 298 – replace ‘osteroradionecrosis’ with ‘osteoradionecrosis’

10.  Line 303 – replace ‘altenative’ with ‘alternative’

11.  Line 325 – add ‘radiotherapy’ after ‘hypofractionated’

Author Response

We appreciate very much your comments to improve our manuscript. Please see the attachment.

Reviewer 2 Report

This is a perspective about immunotherapy and intensity modulated, image-guided radiotherapy for old patients with locally advanced head and neck cancer. The proposal is well described and interesting.

The paper is well written. However, some issues remain.

In chapter 2, the authors described results for non-head and neck cancer. Please add more data about immunotherapy in young head and neck cancer patients.

A table summarizing advantages and disadvantages of immunotherapy and chemotherapy may heal the readers.

In chapter 9, the authors should add more data about immunotherapy and radiotherapy doses for the proposed protocol.

Author Response

(The authors gave the same response as above.)

Round 2

Reviewer 1 Report

All comments were adequately addressed. The scientific content and the readability of the paper have improved.